# Proactive Pseudo-Intervention: Pre-informed Contrastive Learning For Interpretable Vision Models

## Abstract

Deep neural networks excel at comprehending complex visual signals, delivering on par or even superior performance to that of human experts. However, ad-hoc visual explanations of model decisions often reveal an alarming level of reliance on exploiting non-causal visual cues that strongly correlate with the target label in training data. As such, deep neural nets suffer compromised generalization to novel inputs collected from different sources, and the reverse engineering of their decision rules offers limited interpretability. To overcome these limitations, we present a novel contrastive learning strategy called *Proactive Pseudo-Intervention* (PPI) that leverages proactive interventions to guard against image features with no causal relevance. We also devise a novel pre-informed salience mapping module to identify key image pixels to intervene, and show it greatly facilitates model interpretability. To demonstrate the utility of our proposals, we benchmark it on both standard natural images and challenging medical image datasets. PPI-enhanced models consistently deliver superior performance relative to competing solutions, especially on out-of-domain predictions and data integration from heterogeneous sources. Further, saliency maps of models that are trained in our PPI framework are more succinct and meaningful.

## 1 Introduction

Deep neural networks hold great promise in applications requiring the analysis and comprehension of complex imagery. Recent advances in hardware, network architectures, and model optimization, along with the increasing availability of large-scale annotated datasets Krizhevsky et al. (2009); Deng (2012); Deng et al. (2009), have enabled these models to match and sometimes outperform human experts on a number of tasks, including natural image classification Krizhevsky et al. (2017), objection recognition Girshick et al. (2014), disease diagnosis Sajda (2006), and autonomous driving Chen et al. (2015), among others.

While deep learning solutions have been positively recognized for their ability to learn *black-box* models in a purely data driven manner, their very nature makes them less credible for their inability to communicate the reasoning for making predictions in a way that is comprehensible to humans Hooker et al. (2019); Rebuffi et al. (2020). This denies consequential applications where the reliability and trustworthiness of a prediction are of primary concern and require expert audit, *e.g.*, in healthcare Sajda (2006). To stimulate widespread use of deep learning models, a means of interpreting predictions is necessary. However, model interpretation techniques often reveal a concerning fact, that deep learning models tend to assimilate spurious correlations that do not necessarily capture the causal relationship between the input (image) and output (label) Wang et al. (2020a). This issue is particularly notable in small-sample-size (weak supervision) scenarios or when the sources of non-informative variation are overwhelming, thus likely to cause severe overfitting. These can lead to catastrophic failures on deployment Fukui et al. (2019); Wang et al. (2019b).

A growing recognition of the issues associated with the lack of interpretable predictions is well documented in recent years Adebayo et al. (2018); Hooker et al. (2019); Rebuffi et al. (2020). Such phenomenon has energized researchers to actively seek creative solutions. Among these, two streams of work, namely *saliency mapping* Zhao et al. (2018); Simonyan et al. (2013); Dabkowski & Gal (2017) and *causal representation learning* (CRL) Johansson et al. (2016); Wang et al. (2020b); Arjovsky et al. (2019), stand out as some of the

most promising directions. Specifically, saliency mapping encompasses techniques for *post hoc* visualizations on the input (image) space to facilitate the interpretation of model predictions. This is done by projecting the key features used in prediction back to the input space, resulting in the commonly known *saliency maps*. Importantly, these maps do not directly contribute to model learning. Alternatively, CRL solutions are built on the principles of establishing invariance from the data, and it entails teasing out sources of variation that are spuriously associated with the model output (labels). CRL models, while emphasizing the differences between causation and correlation, are not subject to the rigor of causal inference approaches, because their goal is not to obtain accurate causal effect estimates but rather to produce robust models with better generalization ability relative to their naively learned counterparts Arjovsky et al. (2019).

In this work, we present *Proactive Pseudo-Intervention* (PPI), a solution that accounts for the needs of causal representation identification and visual verification. Our key insight is the derivation of pre-informed saliency maps which facilitate visual verification of model predictions *and* enable learning that is robust to (non-causal) associations. While true causation can only be established through experimental interventions, we leverage tools from contrastive representation learning to synthesize pseudo-interventions from observational data. Our procedure is motivated by the causal argument: perturbing the non-causal features will not change the target label.

To motivate, in Figure 1 we present an example to illustrate the benefits of producing causally-informed saliency maps. In this scenario, the task is to classify two bird species (A and B) in the wild. Due to the differences in their natural habitats, A-birds are mostly seen resting on rocks, while B-birds are more commonly found among bulrushes. A deep model, trained naively, will tend to associate the background characteristics with the labels, knowing these strongly correlate with the bird species (labels) in the training set. This is confirmed by the saliency maps derived from the layer-wise relevance propagation (LRP) techniques Bach et al. (2015): the model also attends heavily on the background features, while the difference in bird anatomy is what causally determines the label. If we were provided with an image of a bird in an environment foreign to the images in the training set, the model will be unable to make a reliable prediction, thus causing robustness concerns. This generalization issue worsens with a smaller training sample size. On the other hand, saliency maps from our PPI-enhanced model successfully focus on the bird anatomy, and thus will be robust to environmental changes captured in the input images.

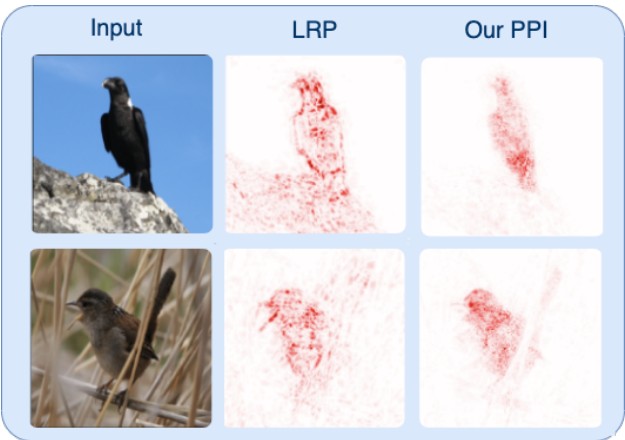

Figure 1: Interpretation for the bird-classification models using saliency maps generated by LRP (*layer-wise relevance propagation*) and our model PPI. LRP shows that naively trained deep model makes decisions based on the background cues (habitat, *e.g.*, rocks, bulrushes) that are spuriously correlated with the bird species, while our pre-informed PPI mostly focuses on the bird anatomy, that generalizes beyond the natural habitat.

PPI addresses causally-informed reasoning, robust learning, and model interpretation in a unified framework. A new saliency mapping method, named *Weight Back Propagation* (WBP), is also proposed to generate more concentrated intervention mask for PPI training. The key contributions of this paper include:

- An end-to-end contrastive representation learning strategy PPI that employs proactive interventions to identify causally relevant features.

- A fast and architecture-agnostic saliency mapping module WBP that delivers better visualization and localization performance.

- Experiments demonstrating significant performance boosts from integrating PPI and WBP relative to competing solutions, especially on out-of-domain predictions, data integration with heterogeneous sources and model interpretation.

## 2 Background

**Visual Explanations.** Saliency mapping collectively refers to a family of techniques to understand and interpret black-box image classification models, such as deep neural networks Adebayo et al. (2018); Hooker et al. (2019); Rebuffi et al. (2020). These methods project the model understanding of the targets, *i.e.*, labels, and their predictions back to the input space, which allows for the visual inspection of automated reasoning and for the communication of predictive visual cues to the user or human expert, aiming to shed model insights or to build trust for deep-learning-based systems.

In this study, we focus on *post hoc* saliency mapping strategies, where saliency maps are constructed given an arbitrary prediction model, as opposed to relying on customized model architectures for interpretable predictions Fukui et al. (2019); Wang et al. (2019b), or to train a separate module to explicitly produce model explanations Fukui et al. (2019); Goyal et al. (2019); Chang et al. (2018); Fong & Vedaldi (2017); Shrikumar et al. (2017). Popular solutions under this category include: activation mapping Zhou et al. (2016); Selvaraju et al. (2017), input sensitivity analysis Shrikumar et al. (2017), and relevance propagation Bach et al. (2015). Activation mapping based methods fail at visualizing fine-grained evidence, which is particularly important in explaining medical classification models Du et al. (2018); Selvaraju et al. (2017); Wagner et al. (2019). Input sensitivity analysis based methods produce fine-grained saliency maps. However, these maps are generally less concentrated Dabkowski & Gal (2017); Fong & Vedaldi (2017) and less interpretable. Relevance propagation based methods, like LRP and its variants, use complex rules to prioritize positive or large relevance, making the saliency maps visually appealing to human. However, our experiments demonstrate that LRP and its variants highlight spuriously correlated features (boarderlines and backgrounds). By contrast, our WBP backpropagates the weights through layers to compute the contributions of each input pixel, which is truly faithful to the model, and WBP tends to highlight the target objects themselves rather than the background. At the same time, the simplicity and efficiency makes WBP easily work with other advanced learning strategies for both model diagnosis and improvements during training.

Our work is in a similar spirit to Fong & Vedaldi (2017); Dabkowski & Gal (2017); Chang et al. (2018); Wagner et al. (2019), where meaningful perturbations have been applied to the image during model training, to improve prediction and facilitate interpretation. Poineering works have relied on user supplied "ground-truth" explainable masks to perturb Ross et al. (2017); Li et al. (2018); Rieger et al. (2020), however such manual annotations are costly and hence rarely available in practice. Alternatively, perturbations can be computed by solving an optimization for each image. Such strategies are costly in practice and also do not effectively block spurious features. Very recently, exploratory effort has been made to leverage tools from counterfactual reasoning Goyal et al. (2019) and causal analysis O'Shaughnessy et al. (2020) to derive visual explanations, but do not lend insights back to model training. Our work represents a fast, principled solution that overcomes the above limitations. It automatically derives explainable masks faithful to the model and data, without explicit supervision from user-generated explanations.

**Contrastive Learning.** There has been growing interest in exploiting contrastive learning (CL) techniques for representations learning Oord et al. (2018); Chen et al. (2020); He et al. (2020); Khosla et al. (2020); Tian et al. (2019). Originally devised for density estimation Gutmann & Hyvärinen (2010), CL exploits the idea of *learning by comparison* to capture the subtle features of data, *i.e.*, positive examples, by contrasting them with negative examples drawn from a carefully crafted noise distribution. These techniques aim to avoid representation collapse, or to promote representation consistency, for downstream tasks. Recent developments, both empirical and theoretical, have connected CL to information-theoretic foundations Tian

et al. (2019); Grill et al. (2020), thus establishing them as a suite of *de facto* solutions for unsupervised representation learning Chen et al. (2020); He et al. (2020).

The basic form of CL is essentially a binary classification task specified to discriminate positive and negative examples. In such a scenario, the binary classifier is known as the critic function. Maximizing the discriminative power wrt the critic and the representation sharpens the feature encoder. Critical to the success of CL is the choice of appropriate noise distribution, where the challenging negatives, *i.e.*, those negatives that are more similar to positive examples, are often considered more effective contrasts. In its more generalized form, CL can naturally repurpose the predictor and loss functions without introducing a new critic Tian et al. (2019). Notably, current CL methods are not immune to spurious associations, a point we wish to improve in this work.

**Causality and Interventions.** From a causality perspective, humans learn via actively interacting with the environment. We intervene and observe changes in the outcome to infer causal dependencies. Machines instead learn from static observations that are unable to inform the structural dependencies for causal decisions. As such, perturbations to the external factors, *e.g.*, surroundings, lighting, viewing angles, may drastically alter machine predictions, while human recognition is less susceptible to such nuisance variations. Such difference is best explained with the *do*-notation Pearl (2009).

Unfortunately, carrying out real interventional studies, *i.e.*, randomized control trials, to intentionally block non-causal associations, is oftentimes not a feasible option for practical considerations, *e.g.*, due to cost and ethics. This work instead advocates the application of synthetic interventions to uncover the underlying causal features from observational data. Specifically, we proactively edit $x$ and its corresponding label $y$ in a data-driven fashion to encourage the model to learn potential causal associations. Our proposal is in line with the growing appreciation for the significance of establishing causality in machine learning models Schölkopf (2019). Via promoting invariance Arjovsky et al. (2019), such causally inspired solutions demonstrate superior robustness to superficial features that do not generalize Wang et al. (2019a). In particular, Suter et al. (2019); Zhang et al. (2020) showed the importance and effectiveness of accounting for interventional perspectives. Our work brings these causal views to construct a simple solution that explicitly optimizes visual interpretation and model robustness.

## 3 Proactive Pseudo-Intervention

Below, we describe the construction of *Proactive Pseudo-Intervention* (PPI), a causally-informed contrastive learning scheme that seeks to simultaneously improve the accuracy, robustness, generalization and interpretability of deep-learning-based computer vision models.

The PPI learning strategy, schematically summarized in Figure 2, consists of three main components: ($i$) a saliency mapping module that highlights causally relevant features; ($ii$) an intervention module that synthesizes contrastive samples; and ($iii$) the prediction module, which is standard in recent vision models, *e.g.*, VGG Simonyan & Zisserman (2014), ResNet He et al. (2016), and Inception Net Szegedy et al. (2016). Motivated by the discussions from our introduction, PPI establishes a feedback loop between the saliency map module and the prediction module, which is interfaced by the synthesized contrastive examples in the intervention module. Under this configuration, the prediction module is encouraged to modify its predictions only when provided with causally-relevant synthetic interventions. Note that components ($i$) and ($ii$) do not involve any additional parameters or neural network modules, which makes our strategy readily applicable to the training of virtually any computer vision task without major customization. Details of these building blocks are given below.

### 3.1 Synthetic causal interventions for contrasts

Key to our formulation is the design of a synthetic intervention strategy that generates contrastive examples to reinforce causal relevance during model training. Given a causal saliency map $s_m(x)$ for an input $x$ wrt label $y = m$, where $m = 1, \ldots, M$, and $M$ is the number of classes, the synthetic intervention consists of removing (replacing with zero) the causal information from $x$ contained in $s_m(x)$, and then using it as the contrastive learning signal.

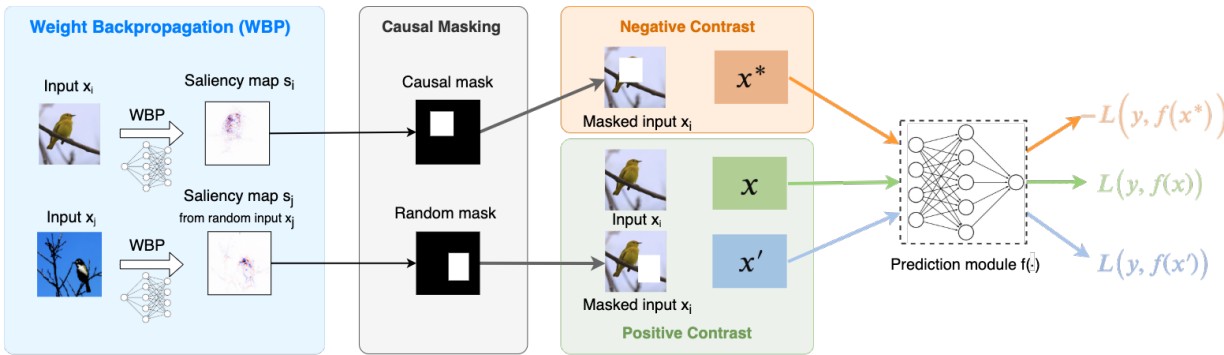

Figure 2: Illustration of the proposed PPI learning strategy. Input images are intervened by removing the saliency map based masks, which alters the input label (*e.g.*, negative control). For positive contrast, we use the original input as well as an input masked with a random slaiency map. We use WBP for the generation of saliency maps.

For now, let us assume the causal salience map $\boldsymbol{s}_m(\boldsymbol{x})$ is known; the procedure to obtain the saliency map will be addressed in the next section. For notational clarity, we use subscript $i$ to denote entities associated with the $i$-th training sample, and omit the dependency on learnable parameters. To remove causal information from $\boldsymbol{x}_i$ and obtain a negative contrast $\boldsymbol{x}_i^*$, we apply the following *soft-masking*

$$\boldsymbol{x}_i^* = \boldsymbol{x}_i - T(\boldsymbol{s}_m(\boldsymbol{x}_i)) \odot \boldsymbol{x}_i, \tag{1}$$

where $T(\cdot)$ is a differentiable masking function and $\odot$ denotes element-wise (Hadamard) multiplication. Specifically, we use the thresholded sigmoid for masking:

$$T(\boldsymbol{s}_m(\boldsymbol{x}_i)) = \frac{1}{1 + \exp(-\omega(\boldsymbol{s}_m(\boldsymbol{x}_i) - \sigma))}, \tag{2}$$

where $\sigma$ and $\omega > 0$ are the threshold and scaling parameters, respectively. We set the scaling $\omega$ so that $T(\boldsymbol{s})$ will result in a sharp transition from 0 to 1 near $\sigma$. Using equation 1 we define the contrastive loss as

$$L_{con}(\theta) = \sum_i \ell(\boldsymbol{x}_i^*, \neg y; f_\theta), \tag{3}$$

where $f_\theta$ is the prediction module, $\ell(\boldsymbol{x}, y; f_\theta)$ is the loss function we wish to optimize, *e.g.* cross entropy, and $\neg$ is used to denote that the original class label has been flipped. In the binary case, $\neg y = 1 - y$, and in the multi-class case it can be interpreted accordingly, *e.g.*, using a one *vs.* others cross-entropy loss. In practice, we set $\ell(\boldsymbol{x}, y; f_\theta) = -\ell(\boldsymbol{x}, y; f_\theta)$. We will show in the experiments that this simple and intuitive causal masking strategy works well in practice (see Tables 2 and 4, and Figure 5). Alternatively, we also consider a *hard-masking* approach in which a minimal bounding box covering the thresholded saliency map is removed. See the Appendix for details.

Note that we are making the implicit assumption that the saliency map is uniquely determined by the prediction module $f_\theta$. While optimizing equation 3 explicitly attempts to improve the fit of the prediction module $f_\theta$, it also implicitly informs the causal saliency mapping. This is sensible because if a prediction is made using non-causal features, which implies the associated saliency map $\boldsymbol{s}_m(\boldsymbol{x})$ is also non-causal, then we should expect that after applying $\boldsymbol{s}_m(\boldsymbol{x})$ to $\boldsymbol{x}$ using equation 1, we can still expect to make the correct prediction, *i.e.*, the true label, for both positive (the original) and negative (the intervened) samples.

**Saliency map regularization.** Note that naively optimizing equation 3 can lead to degenerate solutions for which any saliency map that satisfies the causal sufficiency, *i.e.*, encompassing all causal features, is a valid causal saliency map. For example, a trivial solution where the saliency map covers the entire image may be considered causal. To protect against such degeneracy, we propose to regularize the $L_1$-norm of the saliency map to encourage succinct (sparse) representations, *i.e.*, $L_{reg} = \|\boldsymbol{s}_m\|_1$, for $m = 1, \ldots, M$.

**Adversarial positive contrasts.** Another concern with solely optimizing equation 3 is that models can easily overfit to the intervention, *i.e.*, instead of learning to capture causal relevance, the model learns to predict interventional operations. For example, the model can learn to change its prediction when it detects that the input has been intervened, regardless of whether the image is missing causal features. So motivated, we introduce adversarial positive contrasts:

$$\boldsymbol{x}_i' = \boldsymbol{x}_i - T(\boldsymbol{s}_m(\boldsymbol{x}_j)) \odot \boldsymbol{x}_i, \quad i \neq j, \tag{4}$$

where we intervene with a *false* saliency map, *i.e.*, $\boldsymbol{s}_m(\boldsymbol{x}_j)$ is the saliency map from a different input $\boldsymbol{x}_j$, while still encouraging the model to make the correct prediction via

$$L_{ad}(\theta) = \sum_i \ell(\boldsymbol{x}_i', y; f_\theta), \tag{5}$$

where $\boldsymbol{x}_i'$ is the adversarial positive contrast. The complete loss for the proposed model, $L = L_{cls} + L_{con} + L_{reg} + L_{ad}$, consists of the contrastive loss in equation 3, the regularization loss, $L_{reg}$, and the adversarial loss in equation 5.

## 3.2 Saliency Weight Backpropagation

PPI requires a module to generate saliency maps that inform decision-driving features in the (raw) pixel space. This module is expected to: i) generate high-quality saliency maps that faithfully reflect the model's focus, ii) efficient, as it will be used repeatedly in the PPI training framework. There is a tension between these two goals: advanced saliency maps methods are usually time-consuming Smilkov et al. (2017); Sundararajan et al. (2017). PPI finds a sweet point that better balances the trade-offs: the *Weight Back Propagation* (WBP). WBP is a novel computationally efficient scheme for saliency mapping, and it is applicable to arbitrary neural architectures. Heuristically, WBP evaluates individual contributions from each pixel to the final class-specific prediction. Empirical examinations reveals that WBP results to be more causally-relevant relative to competing solutions based on human judgement (refer to Figure 3 visualization).

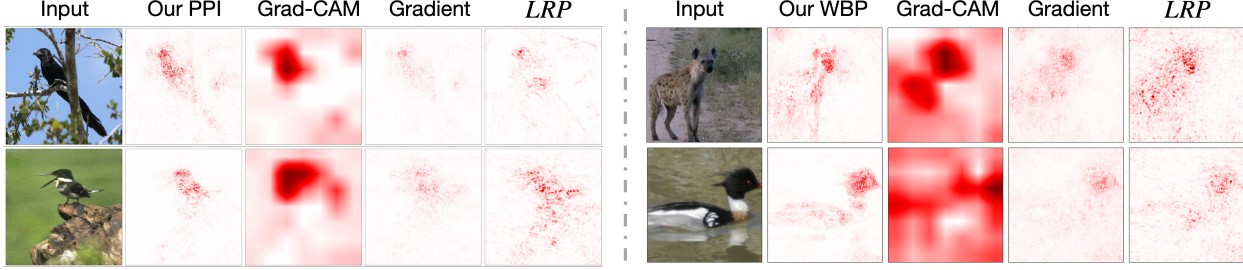

Figure 3: Visualization of the inferred saliency maps. Left: CUB dataset (PPI is based on WBP). Right: ImageNet dataset.

To simplify our presentation, we first consider a vector input and a linear mapping. Let $\boldsymbol{v}^l$ be the internal representation of the data at the $l$-th layer, with $l = 0$ being the input layer, *i.e.*, $\boldsymbol{v}^0 = \boldsymbol{v}$, and $l = L$ being the penultimate *logit* layer prior to the softmax transformation, *i.e.*, $\mathbb{P}(y|\boldsymbol{v}) = \text{softmax}(\boldsymbol{v}^L)$. To assign the relative importance to each hidden unit in the $l$-th layer, we notationally collapse all transformations after $l$ into an operator denoted by $\tilde{\boldsymbol{W}}^l$, which we call the *saliency matrix*, satisfying,

$$\boldsymbol{v}^L = \tilde{\boldsymbol{W}}^l \boldsymbol{v}^l, \quad \forall l \in [0, \dots, L], \tag{6}$$

where $\boldsymbol{v}^L$ is an $M$-dimensional vector corresponding to the $M$ distinct classes in $y$. Though presented in a matrix form in a slight abuse of notation, *i.e.*, the instantiation of the operator $\tilde{\boldsymbol{W}}^l$ effectively depends on the input $\boldsymbol{v}$, thus all nonlinearities have been effectively absorbed into it. We posit that for an object associated with a given label $y = m$, its causal features are subsumed in the interactions between the $m$-th row of $\tilde{\boldsymbol{W}}^0$ and input $\boldsymbol{v}$, *i.e.*,

$$[\boldsymbol{s}_m(\boldsymbol{v})]_k = [\tilde{\boldsymbol{W}}^0]_{mk}[\boldsymbol{v}]_k, \tag{7}$$

where $[\boldsymbol{s}_m(\boldsymbol{v})]_k$ denotes the $k$-th element of the saliency map $\boldsymbol{s}_m(\boldsymbol{v})$ and $[\tilde{\boldsymbol{W}}^0]_{mk}$ is a single element of $\tilde{\boldsymbol{W}}^0$. A key observation for computation of $\tilde{\boldsymbol{W}}^l$ is that it can be done recursively. Specifically, let $g_l(\boldsymbol{v}^l)$ be the transformation at the $l$-th layer, *e.g.*, an affine transformation, convolution, activation, normalization, *etc.*, then it holds that

$$\tilde{\boldsymbol{W}}^{l+1}\boldsymbol{v}^{l+1} = \tilde{\boldsymbol{W}}^{l+1}g_l(\boldsymbol{v}^l) = \tilde{\boldsymbol{W}}^l\boldsymbol{v}^l. \tag{8}$$

This allows for recursive computation of $\tilde{\boldsymbol{W}}^l$ via

$$\tilde{\boldsymbol{W}}^l = G(\tilde{\boldsymbol{W}}^{l+1}, g_l), \quad \tilde{\boldsymbol{W}}^L = 1, \tag{9}$$

where $G(\cdot)$ is the update rule. We list the update rules for common transformations in deep networks in Table 1, with corresponding derivations detailed below.

Table 1: WBP update rules for common transformations.

| Transformation | $G(\cdot)$ |
|---|---|
| *Activation Layer* | $\tilde{\boldsymbol{W}}^l = h \circ \tilde{\boldsymbol{W}}^{l+1}$ |
| *FC Layer* | $\tilde{\boldsymbol{W}}^l = \tilde{\boldsymbol{W}}^{l+1}\boldsymbol{W}^l$ |
| *Convolutional Layer* | $\tilde{\boldsymbol{W}}^l = \tilde{\boldsymbol{W}}^{l+1} \otimes [\boldsymbol{W}^l]_{flip_{2,3}}^{T_{0,1}}$ |
| *BN Layer* | $\tilde{\boldsymbol{W}}^l = \frac{\tilde{\boldsymbol{W}}^{l+1}}{\sigma}\gamma$ |
| *Pooling Layer* | Relocate/Distribute $\tilde{\boldsymbol{W}}^{l+1}$ |

**Fully-connected (FC) layer.** The FC transformation is the most basic operation in deep neural networks. Below we omit the bias term as it does not directly interact with the input. Assuming $g_l(\boldsymbol{v}^l) = \boldsymbol{W}^l\boldsymbol{v}^l$, it is readily seen that

$$\tilde{\boldsymbol{W}}^{l+1}\boldsymbol{v}^{l+1} = \tilde{\boldsymbol{W}}^{l+1}g_l(\boldsymbol{v}^l) = (\tilde{\boldsymbol{W}}^{l+1}\boldsymbol{W}^l)\boldsymbol{v}^l, \tag{10}$$

so $\tilde{\boldsymbol{W}}^l = \tilde{\boldsymbol{W}}^{l+1}W^l$. Graphical illustration with standard affine mapping and ReLU activation can be found in the Appendix.

**Nonlinear activation layer.** Considering that an activation layer simply *rescales* the saliency weight matrices, *i.e.*, $\boldsymbol{v}^{l+1} = g_l(\boldsymbol{v}^l) = h^l \circ \boldsymbol{v}^l$, where $\circ$ is the composition operator, we obtain $\tilde{\boldsymbol{W}}^l = h \circ \tilde{\boldsymbol{W}}^{l+1}$. Using the ReLU activation as a concrete example, we have $h(\boldsymbol{v}^l) = \mathbb{1}\{\boldsymbol{v}^l \geq 0\}$.

**Convolutional layer.** The convolution is a generalized form of linear mapping. In practice, convolutions can be expressed as tensor products of the form $\tilde{\boldsymbol{W}}^l = \tilde{\boldsymbol{W}}^{l+1} \otimes [\boldsymbol{W}^l]_{flip_{2,3}}^{T_{0,1}}$, where $\boldsymbol{W}^l \in \mathbb{R}^{D_2 \times D_1 \times (2S+1) \times (2S+1)}$ is the convolution kernel, $T_{0,1}$ is the transpose in dimensions 0 and 1 and $flip_{2,3}$ is an exchange in dimensions 2 and 3. See the Appendix for details.

**Pooling and normalization layer.** Summarization and standardization are two other essential operations for the success of deep neural networks, achieved by pooling and batch normalization (BN) techniques, respectively. They too can be considered as special instantiations of linear operations. Here we summarize the two most popular operations in Table 1.

## 4 Experiments

To validate the utility of our approach, we consider both natural and medical image datasets, and compare it to existing state-of-the-art solutions. All the experiments are implemented in PyTorch. The source code will be available at `https://github.com/author_name/PPI`. Due to space limitation, details of the experimental setup and additional analyses are deferred to the Appendix.

**Datasets.** We present our findings on five representative datasets: (*i*) `CIFAR-10` Krizhevsky et al. (2009); (*ii*) `ImageNet (ILSVRC2012)` Russakovsky et al. (2015); (*iii*) `CUB` Wah et al. (2011), a natural image dataset with over $12k$ photos for classification of 200 bird species in the wild, heavily confounded by the background characteristics; (*iv*) `GA` Leuschen et al. (2013), a new medical image dataset for the prediction of *geographic atrophy* (GA) using 3D *optical coherence tomography* (OCT) image volumes, characterized by small sample

size (275 subjects) and highly heterogeneous (collected from 4 different facilities); and (*v*) `LIDC-IDRI` Langlotz et al. (2019), a public medical dataset of $1,085$ lung lesion CT images annotated by 4 radiologists. Detailed specifications are described in the Appendix.

**Baselines.** We compare model trained with and without PPI framework to show gains on classification, model interpretability, and cross-domain generalization. Meanwhile, we compare our proposed WBP to the following set of popular saliency mapping schemes: (*i*) Gradient: standard gradient-based salience mapping; (*ii*) GradCAM Selvaraju et al. (2017): gradient-weighted class activation mapping; (*iii*) LRP Bach et al. (2015): layer-wise relevance propagation and its variants. We do no consider more advanced saliency mapping schemes, like perturbation based methods, because it is too time consuming to be used for training purposes.

**Hyperparameters.** The final loss of the proposed model is a weighted summation of four losses: $L = L_{cls} + w_1 L_{con} + w_2 L_{reg} + w_3 L_{ad}$. The weights are simply balanced to match the magnitude of $L_{cls}$, *i.e.*, $w_3 = 1$, $w_2 = 0.1$, and $w_1 = 1$ (CUB, Cifar-10, and GA) and $= 10$ (LIDC). See Appendix Sec B for details about the masking parameters $\sigma$ and $\omega$.

### 4.1 Natural Image Datasets

**Classification Gains.** In this experiment, we investigate how the different pairings of PPI and saliency mapping schemes (*i.e.*, GradCAM, LRP, WBP) affect performance. In Table 2, the first row represents VGG11 model trained with only classification loss, and the following rows represent VGG11 trained with PPI with different saliency mapping schemes. We see consistent performance gains in accuracy via incorporating PPI training on both CUB and CIFAR-10 datasets. The gains are mostly significant when using our WBP for saliency mapping (improving the accuracy from 0.662 to 0.696 on CUB, and from 0.881 to 0.901 on CIFAR-10).

Table 2: Performance improvements achieved by training with PPI on CUB, CIFAR-10, and GA dataset. We report means and standard deviations (SDs) from 5-fold cross-validation for GA prediction.

| Models | CUB (Acc) | Cifar-10 (Acc) | GA (AUC $\pm$ SD) |
|---|---|---|---|
| Classification | 0.662 | 0.881 | $0.877 \pm 0.040$ |
| +PPI$_{Gradient}$ | 0.673 | 0.885 | $0.890 \pm 0.035$ |
| +PPI$_{LRP}$ | 0.680 | 0.891 | $0.895 \pm 0.037$ |
| +PPI$_{GradCAM}$ | 0.683 | 0.895 | $0.908 \pm 0.036$ |
| +PPI$_{WBP}$ | **0.696** | **0.901** | $\mathbf{0.925 \pm 0.023}$ |

**Model Interpretability.** In this task, we want to qualitatively and quantitatively compare the causal relevance of saliency maps generated by our proposed model and its competitors. In Figure 3(right), we show the saliency maps produced by different approaches for a VGG11 model trained on CUB. Visually, gradient-based solutions (Grad and GradCAM) tend to yield overly dispersed maps, indicating a lack of specificity. LRP gives more appealing saliency maps. However, these maps also heavily attend to the spurious background cues that presumably help with predictions. When trained with PPI (based on WBP), the saliency maps focus on the causal related pixels, i.e., special parts of birds body.

To quantitatively evaluate the causal relevance of competing saliency maps, we adopt the evaluation scheme proposed in Hooker et al. (2019), consisting of masking out the contributing saliency pixels and then calculating the reduction in prediction score. A larger reduction is considered better for accurately capturing the pixels that 'cause' the prediction. Results are summarized in Figure 4a, where we progressively remove the top-$k$ saliency points, with $k = 100, 500, 1000, 5000, 10000$ ($10000 \approx 6.6\%$ of all pixels), from the CUB test input images. Our PPI consistently outperforms its counterparts, with its lead being most substantial in the low-$k$ regime. Notably, for large $k$, PPI removes nearly all predictive signal. This implies PPI specifically targets the causal features. Quantitative evaluation with additional metrics are provided in the Appendix.

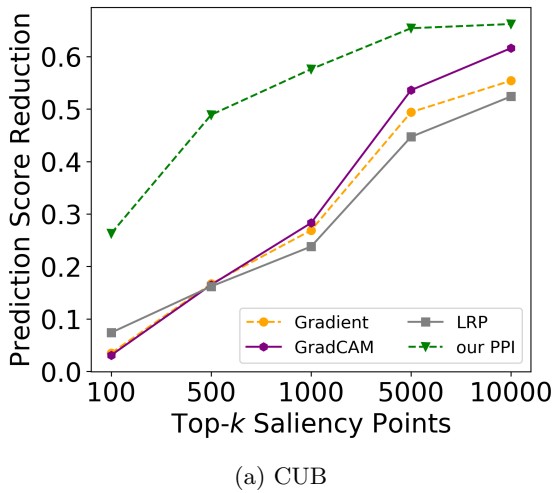
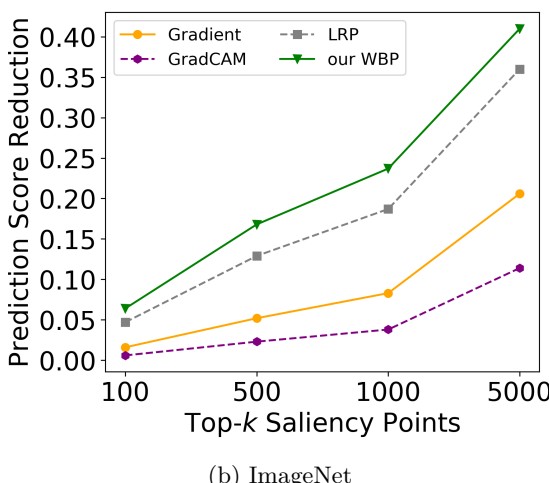

(a) CUB

(b) ImageNet

Figure 4: Quantitative evaluations of causal relevance of competing saliency maps (higher is better).

To test the performance of WBP itself (without being trained with PPI), we compare WBP with different approaches for a VGG11 model trained on ImageNet from PyTorch model zoo. Figure 3(left) shows that saliency maps generated by WBP more concentrate on objects themselves. Also, thanks to the fine resolution of WBP, the model pays more attention to the patterns on the fur to identify the leopard (row 1). This is more visually consistent with human judgement. Figure 4b demonstrates WBP identifies more causal pixels on ImageNet validation images.

## 4.2 OCT-GA: Geographic Atrophy Classification

Next we show how the proposed PPI handles the challenges of small training data and heterogeneity in medical image datasets. In this experiment (with our new dataset, that we will make public), each OCT volume image consists of 100 scans of a $512 \times 1000$ sized image Boyer et al. (2017). We use a multi-view CNN model Su et al. (2015) to process such 3D OCT inputs, and use it as our baseline solution (see the Appendix). We investigate how the different saliency mapping schemes (*i.e.*, Grad, GradCAM, LRP, WBP) work with PPI. For WBP, we also tested the bounding box variant, denoted as WBP (box) (see the Appendix). In Table 2, we see consistent performance gains in AUC score via incorporating PPI training (from 0.877 to 0.925, can be improve to 0.937 by PPI with WBP(box)), accompanied by the reductions in model variation evaluated by the standard deviations of AUC from the five-fold cross-validation. The gains are most significant when using our WBP for saliency mapping. The model trained by PPI with WBP is significantly different with other baseline models based on the Delong test for receiving operating characteristic comparisons DeLong et al. (1988). We further compare the saliency maps generated by these different combinations. We see that without the additional supervision from PPI, competing solutions like Grad, GradCAM and LRP sometimes yield non-sensible saliency maps (attending to image corners). Overall, PPI encourages more concentrated and less noisy saliency maps. Also, different PPI-based saliency maps agree with each other to a larger extent. Our findings are also verified by experts (co-authors, who are ophthalmologists specializing in GA) confirming that the PPI-based saliency maps are clinically relevant by focusing on retinal layers likely to contain abnormalities or lesions. These results underscore the practical value of the proposed proactive interventions.

**Cross-domain generalization.** Common to medical image applications is that training samples are usually integrated from a number of healthcare facilities (*i.e.*, domains), and that predictions are sometimes to be made on subjects at other facilities. Despite big efforts to standardize the image collection protocols, with different imaging systems operated by technicians with varying skills, apparent domain shifts are likely to compromise the cross-domain performance of these models. We show this phenomenon on the `GA` dataset in Table 3, where source samples are collected from four different hospitals in different health systems (A, B, C

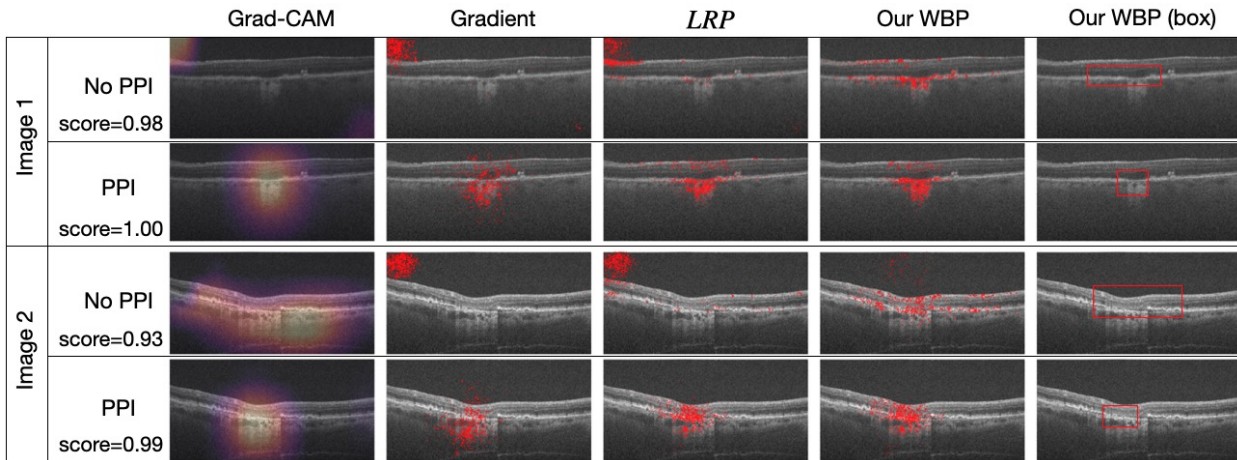

Figure 5: Saliency maps on GA dataset based on models trained with PPI and without PPI. Maps of models trained with PPI are more clinically relevant by focusing on retinal layers likely to contain abnormalities or lesions, and more concentrated.

and D, see the Appendix for details). Each cell contains the AUC of the model trained on site X (row) and tested on site Y (column), with same-site predictions made on hold-out samples. A significant performance drop is observed for cross-domain predictions (off-diagonals) compared to in-domain predictions (diagonals). With the application of PPI, the performance gaps between in-domain and cross-domain predictions are considerably reduced. The overall accuracy gains of PPI further justify the utility of causally-inspired modeling. Notably, site D manifests strong spurious correlation that help in-domain prediction but degrades out-of-site generalization, which is partly resolved by the proposed PPI.

Table 3: AUC results for GA prediction with or without PPI. Models are trained on one site and cross-validated on the other sites. Darker color indicates better performance.

| With PPI | A | B | C | D | Mean | STD |
|---|---|---|---|---|---|---|
| A | 1.000 | 0.906 | 0.877 | 0.865 | **0.912** | **0.061** |
| B | 0.851 | 0.975 | 0.863 | 0.910 | **0.900** | 0.056 |
| C | 0.954 | 0.875 | 0.904 | 0.931 | **0.916** | **0.034** |
| D | 0.824 | 0.846 | 0.853 | 0.904 | **0.857** | **0.034** |
| No PPI | A | B | C | D | Mean | STD |
| A | 1.000 | 0.854 | 0.832 | 0.827 | 0.878 | 0.082 |
| B | 0.810 | 0.874 | 0.850 | 0.906 | 0.860 | **0.040** |
| C | 0.860 | 0.779 | 0.873 | 0.862 | 0.843 | 0.043 |
| D | 0.748 | 0.792 | 0.836 | 0.961 | 0.834 | 0.092 |

### 4.3 LIDC-IDRI: Lung Lesions Classification

To further examine the practical advantages of the proposed PPI in real-world applications, we benchmark its utility on LIDC-IDRI; a public lung CT scan dataset Armato III et al. (2011). We followed the preprocessing steps outlined in Kohl et al. (2018) to prepare the data, and adopted the experimental setup from Selvan & Dam (2020) to predict lesions. We use Inception_v3 Szegedy et al. (2016) as our base model for both standard classification and PPI-enhanced training with various saliency mapping schemes. See the Appendix for details.

**Lesion classification.** We first compare PPI to other specialized SOTA network architectures. Table 4 summarizes AUC scores of Tensor Net-X Efthymiou et al. (2019), DenseNet Huang et al. (2017), LoTeNet Selvan

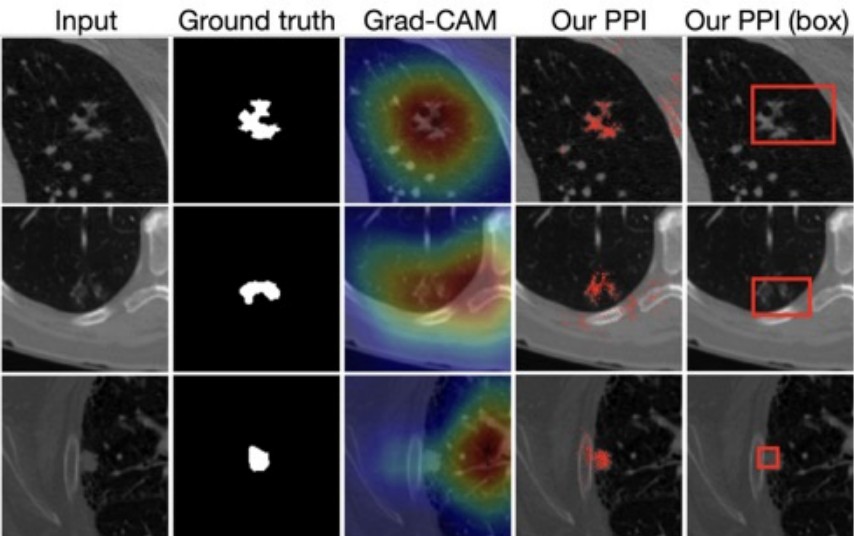

Figure 6: Saliency maps on LIDC-IDR. Saliency maps of PPI+WBP are mostly consistent with the ground truths.

Table 4: LIDC-IDRI classification AUC results.

| Models | AUC |
| --- | --- |
| Tensor Net-X Efthymiou et al. (2019) | 0.823 |
| DenseNet Huang et al. (2017) | 0.829 |
| LoTeNet Selvan & Dam (2020) | 0.874 |
| Inception_v3 Szegedy et al. (2016) | 0.921 |
| +PPI$_{GradCAM}$ | 0.933 |
| +PPI$_{Gradient}$ | 0.930 |
| +PPI$_{LRP}$ | 0.931 |
| +PPI$_{WBP}$ | 0.935 |
| +PPI$_{WBP(box)}$ | **0.941** |

& Dam (2020), Inception_v3 Szegedy et al. (2016), as well as our Inception_v3 trained with and without PPI$_{WBP}$. The proposed PPI$_{WBP(box)}$ leads the performance chart by a considerable margin, improving Inception_v3 from 0.92 to 0.94.

**Weakly-supervised image segmentation.** In Figure 6, we compare saliency maps generated by Grad-CAM, WBP, WBP (box) to the ground truth lesion masks from expert annotations. Note that we have only supplied patch-label labels during training, not the pixel-level expert segmentation masks, which constitute a challenging task of weakly-supervised image segmentation. In line with the observations from the GA experiment, our PPI-training enhanced WBP saliency maps are mostly consistent with the expert segmentations. Together with Table 4, Figure 6 confirms that the proposed PPI+WBP improves both the classification performance and model interpretability.

## 5 Conclusions

We have presented *Proactive Pseudo-Intervention* (PPI), a novel interpretable computer vision framework that organically integrates saliency mapping, causal reasoning, synthetic intervention and contrastive learning. PPI couples saliency mapping with contrastive training by creating artificially intervened negative samples absent of causal features. To communicate model insights and facilitate pre-informed reasoning, we

derived an architecture-agnostic saliency mapping scheme called *Weight Back Propagation* (WBP), which faithfully captures the causally-relevant pixels/features for model prediction. Visual inspections of the saliency maps show that WBP, is more robust to spurious features compared to competing approaches. Empirical results on natural and medical datasets verify the combination of PPI and WBP consistently delivers performance boosts across a wide range of tasks relative to competing solutions, and the gains are most significant where the application is complicated by small sample size, data heterogeneity, or confounded with spurious correlations.

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
