# OpenReview forum: "Proactive Pseudo-Intervention: Pre-informed Contrastive Learning For Interpretable Vision Models"
_TMLR — Rejected by TMLR_

### Review · Reviewer_6PLv · 2022-09-12

**Summary Of Contributions:**

While saliency methods are usually applied to already trained networks (post-hoc), this work proposes to use them during training to mask the training images. In doing so, two challenges arise. First, the saliency map could degenerate (e.g., scoring the entire image important would not mask the image). This is solved with an L1-loss. Second, the model could overfit to the interventions; this is solved by introducing adversarial samples saliency maps.

In addition, this work also proposes a new saliency method, "Saliency Weight Backpropagation". Experiments are conducted on four image datasets against three different baselines. The experiments investigate the classification performance, how the performance decreases when the most salient input pixels are removed, and agreement with ground truth bonding box annotations.



**Broader Impact Concerns:**

I do not have any broader impact concerns.

**Requested Changes:**

Critical for recommendation:

1. Please list the differences between WBP and the Gradient. Adding the pseudocode of WBP would be good.

   If my suspicion is correct that it is equivalent to the Gradient, I would ask to remove the WBP part.

3. Which LRP rule is used?
4. Please be explicit about which combinations of model and saliency method are evaluated. I would recommend labeling methods such as Vanilla+LRP or PPI+LRP to clarify whether the underlying model weights differ.

Strengthen the work:

1. Addressing the weaknesses 3.2, 3.3.
2. Motivate the introduction of WBP better. Why is another method needed?

**Strengths And Weaknesses:**

Strengths:

- The idea of using interventions during the training process is interesting.
- The PPI+WBP method shows an improvement in classification accuracy on three different datasets (see Table 2)
- When removing the most salient pixels, the prediction performance decreases for PPI faster than for the baselines.
-  The evaluation is done on four different datasets.
- The paper is  written clearly, also some motivation is missing (see Weaknesses 3.1-3.3,  2)

Weaknesses:

1. **Baselines**:
   1. I find the selection of only 3 baseline methods rather weak. Many more saliency methods exist, and it would be interesting to see how PPI+WBP performs on saliency methods based on deletion.
   2. LRP has known issues, e.g., some LRP rules do not pass the weight randomization sanity checks. Unfortunately, it is not mentioned which LRP rule is used.
2. **WBP:** A wide variety of saliency methods exists: Why is another method needed? The description of WBP reminds me of the Gradient. What is the difference between WBP and the Gradient? For example,
   1. Assuming $\tilde W^{l+1}$ would correspond to the gradient: $\tilde W^{l+1} = \frac{\partial v^L}{\partial v^{l+1}}.$ Then we would have  $\frac{\partial v^L}{\partial v^{l}} = \frac{\partial v^{l+1}}{\partial v^{l}} \frac{\partial v^L}{\partial v^{l+1}} = \tilde W^{l+1} W^l$  (Using the [numerator layout](https://www.wikiwand.com/en/Matrix_calculus#/Derivatives_with_vectors))
   2. Another question would be: why are the WBP saliency maps positive? The weight matrices W^l Could also introduce negative values.
3. **Missing information**: Throughout the paper, I found many essential details to be missing:
   1. In Figure 4ab: Are the baselines applied to the same model? Or are LRP, Gradient, and GradCAM applied to a different model not trained with PPI?
   2. How are the saliency maps normalized? Why are the Gradient saliency maps positive?
   3. How are the four losses tuned? Are they added together, or is each weighted?

To summarize my review, I like the idea of using the saliency maps for training a neural network, and the empirical results show an improvement in various metrics. However, I want the authors to comment on the differences between their saliency method WBP and provide more information about the baselines (see also next section).

---

> ### Author Response · Authors · 2022-09-27
> **Reply to Reviewer 6PLv**
>
> Thanks to the reviewer for the valuable feedback. We appreciate that the reviewer recognizes our idea interesting and the benefits of our proposed approach. We will address the weakness thoroughly and comment on the requested changes.
>
> ### Baselines 1.1
> The three baseline methods are selected based on the simplicity of applying such methods during the training. The reviewer is right in that there are many other saliency methods. However, perturbation-based methods (Appendix C.1) require interactions for a single saliency map, and other methods that are improved upon the selected baselines (Score-CAM, Grad-CAM++, DeepLift, etc.) require additional computation which is not desired during the long-time training stage. The proposed WBP is a single-pass computational process like the selected baseline methods.
>
> ### Baselines 1.2
> The reviewer points out the instability of the LRP rule. We encountered the same problem implementing LRP on some ResNet networks. For the VGG network, we follow the [tutorial](https://git.tu-berlin.de/gmontavon/lrp-tutorial) that for high-level and mid-level layers the  $\epsilon$ rule is applied ($\epsilon=1e-9$), and for low-level layers, the \gamma rule is applied ($\gamma=0.25$) and at the input layer, the $Z^\beta$ rule is applied. For the Inception-V3 network, we apply the \epsilon rule for all layers for simplicity and without introducing more parameters, and at the input stage, the $Z^\beta$ rule is applied.
>
> ## WBP
> The reviewer asks for more explanation on the difference between WBP and existing saliency methods, especially the gradient method. In short, WBP computes the weight that is analytically applied on each pixel to predict a class while other methods include gradient more or less during the computation (gradient, Grad-CAM, $Z^\beta$ rule in LRP). The downside of including the gradient is the inclusion of spurious explanations when the model struggles to improve the prediction score [1] (also see Figure 5). Computing the weights only will yield consistent explanations even when the model is not improving its confidence in a given class.
>
> ### WBP 2.1
> The computation of gradients follows the chain rule throughout the network layers. The computation of WBP also follows the chain rule throughout the network layers. This may be the reason the formula is similar to the computation of the gradient (note that LRP also follows such a rule). However, instead of passing a gradient, WBP passes a weight matrix backward so that the value of the weight indicates the importance of the pixel when multiplied by the pixel value. Whereas the gradient only provides a level of significance on its own and is not stable when the model struggles to improve its confidence [1].
>
> ### WBP 2.2
> The reviewer is right that WBP results in both positive and negative values. So do all other saliency mapping methods. We are not sure where the reviewer is referencing the positivity of the saliency map. For visualizations, we normalize all saliency maps to a range[0,1] so that a darker red region indicates high importance and a lighter red region indicates low importance. During the training, all saliency maps are masked via $T(.)$ based on Equation (2).
>
> ### Missing information 3.1
> For the deletion metric, all saliency methods are computed on a pre-trained network. So in Figure 4a, the legend should be WBP bot PPI. And in accordance with the reviewer’s suggestion, all saliency methods in Figure 4ab should be vanilla versions.
>
> ### Missing information 3.2
> For visualizations, we normalize all saliency maps to a range[0,1] so that a darker red region indicates high importance and a lighter red region indicates low importance. During the training, all saliency maps are masked via T(.) based on Equation (2).
>
> ### Missing information 3.3
> In section 4. Experiments Hyperparameters, the total loss is the weighted sum of the four losses. And the setting for each loss is listed in the same section.
>
>
> # Reference
> [1] Linardatos, Pantelis, Vasilis Papastefanopoulos, and Sotiris Kotsiantis. "Explainable ai: A review of machine learning interpretability methods." Entropy 23.1 (2020): 18.

---

> > ### Comment · Reviewer_6PLv · 2022-09-27
> > **Re: Rebuttal - Similarity between WBP and Jacobian**
> >
> >
> >
> > First, I want to thank the authors for their detailed comments which addressed most of my questions well.
> >
> > I have an **important** follow-up question regrading the similarity between the gradient and WBP. This might be partially my fault as I should have said Jacobian instead of Gradient:
> >
> > I claim that the operator $\tilde W^{l}$ is equal to the Jacobian of $v^L$ w.r.t $v^{l}$, i.e.:
> >
> > $\tilde W^{l} = \frac{\partial v^L}{\partial v^{l}}$.
> >
> > As argued before the derived rules from Table 1, correspond exactly to the Jacobian rules. For example, for linear layer:
> >
> >  $\frac{\partial v^L}{\partial v^{l}} = \frac{\partial v^{l+1}}{\partial v^{l}} \frac{\partial v^L}{\partial v^{l+1}} = \tilde W^{l+1}W^l  $
> >
> > Or for the activation layer:
> >
> >  $\frac{\partial v^L}{\partial v^{l}} = \frac{\partial g_l(v^{l})}{\partial v^{l}} \frac{\partial v^L}{\partial v^{l+1}} = h(v^l)\circ \tilde W^{l+1} $
> >
> > And also for the final layer it works out:
> >
> > $ \frac{\partial v^L}{\partial v^{L}} = \tilde W^L = 1$
> >
> > Please, either disprove me on the equality between $\tilde W^l$ and the Jacobian or update the manuscript accordingly.
> >
> > I will read the other reviews and rebuttals and might come back with further questions.

---

> ### Author Response · Authors · 2022-09-27
> **Continue Reply to Reviewer 6PLv**
>
> ## Comments on requested changes
> ### Critical for recommendation
> - The difference between WBP and other saliency mapping methods and especially the gradient is explained in the WBP section. We will include the explanation for better clarity.
> - The detailed implementation of LRP is listed in Baseline 1.2. We will include this in the Appendix.
> - Only in the deletion metric, the vanilla model + saliency mapping method is evaluated. In the prediction metric, the PPI+saliency mapping method is evaluated. We will adapt the notation suggested by the reviewer.
> ## Strengthen the work
> - Addressed in 3.2 and 3.3
> - One important motivation for the WBP is the instability of gradient information which leads to spurious cues in predicting a class. Especially for medical images where the model can predict a class with high confidence, the explanation may not be reliable since the gradient only highlights the sensitive region instead of the important region. In contrast, WBP computes analytically the weight for how much each pixel contributes to a prediction and this weight is consistent for a reliable model. We will clarify the motivation for WBP in the paper.

---

### Review · Reviewer_MiEG · 2022-09-14

**Summary Of Contributions:**

# Summary of contributions

The authors present a framework termed Proactive pseudo-intervention, in which a classifier is regularized with 3 losses based on saliency:

1. minimising the L1 loss of the saliency map (favouring sparse input sensitivity)
2. minimising the likelihood of the model predicting class c_i if presented with an input x_i' where the most salient information w.r.t to c_i has been removed from x_i
3. maxinimisng the likelihod of predicting the likelihood of class c_i when presented an x_i' from which the salient information for a *differen3* class c_j has been removed

In order to create the saliency maps the authors present an approach called Weight Backpropagation (WBP), which strongly resembles backpropagation, possibly being viewable as a recurrent layerwise inverse applied to a 1-vector shaped like the logits.

The authors then evaluate the framework in terms of it's generalisation capability on classical CV datasets on CUB,CIFAR-10 and Imagenet as well as 2 medical datasets on CT images of lung lesions and OCT images of geographic atrophy, mostly following the ROAR (Hooker et al. 2019) method and pairing PPI with WBP, vanilla Gradients, GradCAM and LRP, using Densnet,TensorNet-X,LoTeNet and Inception_v3..

**Broader Impact Concerns:**

Give the experiments used medical images  I think a broader impact statement might be appropriate, or at least one could highlight a paper that explores the broader impact of interpretable models and attribution methods in medical applocations in the conclusion ( I think the literature review covers it, but it might get lost on people skimming the work).

**Requested Changes:**

I'm breaking down into *requested* changes which I think *should* be done before acceptance and  *nice to haves*.

# Requested

1. Please make sure you report mean and standard deviation for all tables and perform appropriate statistical significance tests (Bonferroni correction and all those nice tools we have at our disposal now) to make sure the reported changes are actually significant
2. Please address the question of "training on ROAR" in the text, explore the impact and *if possible* suggest possible alternative checks (as a nice to have you can also *run them*). "Training on the benchmark" might actually be totally fine and not *need* further checks but I think it deserves careful consideration and discussion (e.g., how about OOD data? This might be covered by the crossvalidation, but still)
3. Please Justify the model selection and/or add "standard" CV models like resnet variants, efficientnet etc. A justification can be that they are standard in the medical community or they were used in the works that are compared to.
4.  Please make clearer the difference between backpropagation and your method, ideally with more detailed numerical example (e.g. go through ReLU, convolution etc. to show the difference of applying your update rule and simply treating the saliency as a gradient backpropagating a 1 ).

# Nice to have


1. More of a suggestion: I personally understood your method as applying a layerwise inverse to a saliency vector which preserves the total "saliency mass". If this is a correct interpretation, maybe this might replace the gradient-esque explanation currently in the paper?
2. As mentioned in requested, actually *running* additional probes you can come up with would strongly improve the paper, but is not necessesary
3. If you have the computational resources, please run ablations over your hyperparameters and/or justify your choices.
4. Please add a model-card to your PPI release and tally the carbon and compute  expenditure of the methods if possible (ala https://mlco2.github.io/impact/#publish )

**Strengths And Weaknesses:**

Strengths:

1. The WBP method, if I understood it correctly is simple and makes intuitive sense (I'm in fact surprised nobody, including me thought about it before).
2. The evaluation seems to follow best practices of interpretable methods and is done on relevant datasets
3. The writing is *mostly* clear and readable

Weaknesses:

1. *Some* ambiguities in the paper make it a bit hard to call it clear and readable ((see below).
2. The architectures chosen are a bit old (2 years is a long time in ML) and specific to the medial community, which can be okay but should maybe be justified in the text.
3.  Why no means and standard deviations on all results?
4. The related work section should possibly include causal explainability method like https://proceedings.neurips.cc/paper/2019/hash/3ab6be46e1d6b21d59a3c3a0b9d0f6ef-Abstract.html
5. If ROAR is followed exactly, then in a sense the PPI model is "training" on the ROAR framework (as the network is trained to reduce the correct class when the attributed information is removed). This is "fine" but deserves more discussion and maybe probing.

---

Ambiguities:

The most important ambiguities to me are

1. Is the ROAR method followed exactly including *retraining*? In the text it states "consisting of masking out the contributing saliency pixels and then calculating the reduction in predictions score", but it does not mention "retraining on the modified dataset".
2. I think the difference between weight backpropagation and normal backpropagation using Jacobians needs to be made clearer. I think there *is* a difference, but only after carefully thinking about it in terms of accounting.

---

> ### Author Response · Authors · 2022-09-27
> **Reply to Reviewer MiEG**
>
> Thanks to the reviewer for the valuable feedback. We appreciate that the reviewer recognizes that WBP is simple and intuitive, the evaluation is conducted in accordance with the best practice, and the paper is clear in most sections. We will address the weaknesses and ambiguities one by one, and comment on the requested and nice-to-have changes.
> ### Weakness 1
> We will address the ambiguities in the later sections.
> ### Weakness 2
> The reviewer is right that VGG and Inception-V3 (or even ResNet variants) are relatively old models in the CV community. We make decisions on the model selection based on how simple it is to apply different saliency mapping methods to the model architecture and how well these methods explain the predictions. During the implementation, we found that the selected models were simplest to adopt the saliency mapping methods during training. In the medical imaging community, selected models are more widely adopted than the more lately proposed models such as transformers. But in our opinion, PPI and WBP can be applied to all CV model variants.
> ### Weakness 3
> We provide mean and std on the medical dataset GA in Table 2 using 5-fold cross-validation. For other datasets, we can run multiple experiments with other random seeds to evaluate the significance of each method. However, since other datasets have designed training and testing data splits, running multiple random-seeded experiments can result in variance due to random noise variance but this variance should be less significant compared to the improvement of the proposed method. We can run experiments on the natural image dataset to verify if needed.
> ### Weakness 4
> The reviewer suggests including a causal explaining method in the related work. Causal explainability methods ([1][2][3]) explain the prediction of predictive models using a causality graph and/or causal interventions. These post-hoc methods usually require numerous observed data to validate the causal effect and hence deliver an explanation. Therefore, even though a reliable explanation can be obtained, it is harder to adapt such computationally expensive methods during a training process.
> ### Weakness 5
> The reviewer is right in that we evaluate the effectiveness of WBP in selecting important pixels by Removing the pixels according to the assigned importance score. But we only conduct the Removing of ROAR because the main idea is to apply any saliency mapping method to generate factual and counterfactual masks so that the model is more robust to these distortions and spurious relations. Even though Retraining is not done explicitly, L_con and L_adv are ‘retraining’ the model to perturbed image examples. To some extent, the improvement of PPI suggests the benefit of adapting the model to this perturbed dataset via different saliency mapping methods.
> ### Requested 1
> We can report the mean and std for the natural image dataset if the reviewer does not agree that the variance caused by the noise in the randomness does not exceed the improvement of the proposed method (as illustrated in Table 2 GA dataset).
> ### Requested 2
> See Weakness 5 for considerations on ROAR. We only evaluate WBP on the deletion metric to demonstrate that WBP identifies the most important pixels compared to other baselines.
> ### Requested 3
> See Weakness 2 for the reason why we select VGG, inception-V3, and others as our model selections. For ResNet specifically, LRP is not stable in adapting the architecture. We can provide evaluations on ResNet excluding the LRP method.
> ### Requested 4
> The main difference between WBP and gradient is that only the weight is back propagated in WBP. The gradient can be less reliable when the model struggles to improve the prediction and points to spurious cues in the input image. However, the weight associated with the most important pixel is consistent throughout the training. We can provide a simple WBP example of linear layer and ReLU in the appendix. For a convolution layer, a detailed derivation is listed in Appendix A.
>
> # Reference
> [1] Lin, Wanyu, Hao Lan, and Baochun Li. "Generative causal explanations for graph neural networks." International Conference on Machine Learning. PMLR, 2021.
>
> [2] Schwab, Patrick, and Walter Karlen. "Cxplain: Causal explanations for model interpretation under uncertainty." Advances in Neural Information Processing Systems 32 (2019).
>
> [3] Slack, Dylan, et al. "Reliable post hoc explanations: Modeling uncertainty in explainability." Advances in Neural Information Processing Systems 34 (2021): 9391-9404.

---

> > ### Comment · Reviewer_MiEG · 2022-09-27
> > **Subset: On ROAR**
> >
> > Dear Authors,
> >
> > apologies for possible bluntness, as I am squeezing in this reply during the ICLR deadline.
> >
> > If I can summarize your comments: no, you did not apply the full ROAR framework, rather you simply remove pixels from the input. If I'm wrong please correct me (and my apologies for misunderstanding), if I'm correct then please either apply ROAR or modify the text to not make any connection to the framework, as the very paper shows the importance of retraining for proper judgement.  Your statement that "it removes the most important pixels" is in need to justification that your method of evaluation actually determines pixel importance. ROAR achieves this by *failing to retrain after removal*. If you do not retrain, you need a new justification.
> >
> > I will comment on the other points as I find time.

---

> > > ### Author Response · Authors · 2022-10-04
> > > **Reply on ROAR**
> > >
> > > Thanks for clarifying your comments. Our statement that "WBP removes the most important pixel" is based on the deletion metric which is to evaluate how important each pixel identified by a saliency mapping method is to the prediction. Since removing the pixels identified by WBP decreases the confidence score most (shown in Figure 4), we conclude that "WBP removes the most important pixels". We do not think that there is any obvious connection to ROAR in our text since our goal is to learn better interpretable models not to evaluate the interpretability of the model. Our intuition is using the identified pixels to generate positive and negative examples and train the model to be more robust to irrelavant cues. Retraining after removing the pixels multiple times is not efficient and cannot improve the interpretibility of the model. We will include the justification for not retraining in the method section.

---

> > > > ### Comment · Reviewer_MiEG · 2022-10-05
> > > > **There might be a confusion about why I'm asking for retraining**
> > > >
> > > > Dear authors: I apologize if I expressed myself unclearly, but "it is not efficient" misses the point of ROAR. Quoting from the paper
> > > >
> > > > >The replacement value c can only be considered uninformative if the model is trained to learn it as
> > > > such. Without retraining, it is unclear whether degradation in performance is due to the introduction of
> > > > artifacts outside of the original training distribution or because we actually removed information. This
> > > > is made explicit in our experiment in Section 4.3.1, we show that without retraining the degradation
> > > > is far higher than the modest decrease in performance observed with re-training. This suggests
> > > > retraining has better controlled for artefacts introduced by the modification.
> > > >
> > > > Meaning that your evaluation is missing a critical component. We cannot distinguish between a case where WBP is a better attribution method and the case where training with WBP in your PPI framework is simply making the network exceedingly dependent towards the features identified, but actually could learn to compensate (meaning the features aren't actually the most relevant features in the data for the network)

---

> > > > > ### Comment · Reviewer_MiEG · 2022-10-05
> > > > > **Please clarify your notation/WBP (see also reviewer  6PLv)**
> > > > >
> > > > > Dear authors,
> > > > >
> > > > > the title says it all. An easy counter way to explain this would be to take a simple tiny DNN (1 conv layer with kernel size 2 or 3, max pool,1 linear layer, 2 classes, relu) and show step by step the difference between jacobian backprop and your method.

---

> > > > > > ### Comment · Reviewer_MiEG · 2022-10-05
> > > > > > **Please show results on CIFAR10**
> > > > > >
> > > > > > Dear authors,
> > > > > >
> > > > > > since you state that it is easy to apply the method to every type of architecture, I invite you to show results on CIFAR10, resnet18 or   DavidNet https://medium.com/fenwicks/tutorial-2-94-accuracy-on-cifar10-in-2-minutes-7b5aaecd9cdd or the architecture in https://myrtle.ai/how-to-train-your-resnet-8-bag-of-tricks/. Both are optimised to be fast to train on little compute, so this should be an easy experiment to add. Otherwise it is difficult to relate your results to standard ML CV results

---

> ### Author Response · Authors · 2022-09-27
> **Continue Reply to Reviewer MiEG**
>
> ### Nice to have
> - The reviewer is right that WBP like LRP preserves the total saliency mass/relevance through layers of the model. We show WBP in form of gradient-like derivations since this is the simplest way to illustrate the computation of WBP. We can add an additional section to rephrase the interpretation if this helps the audience to further understand our proposed WBP.
> - Running other experiments to evaluate WBP can be nice. The main focus of our paper is applying different saliency mapping during training so that the model interpretation is more reliable. Spending too much effort on validating WBP may distract the focus.
> - We have the requested ablations on GA which we can include in the appendix.
> - We can include the environmental impact if the issue is raised by other reviews too.

---

### Review · Reviewer_x2bF · 2022-09-18

**Summary Of Contributions:**

The authors propose a new training strategy, called Pro-active Pseudo Intervention (PPI), aimed at making sure that deep neural networks rely on causally relevant features to make their predictions, and do not pick on spurious correlation. Through extensive experiments on benchmark, and real datasets, they show that PPI improves the classification performance. In a real medical dataset, and in a practical relevant setting of generalizing across data obtained from multiple hospitals/health systems, the authors show that PPI improves the generalization ability across these hospitals/health systems too. In addition to the improvement in performance, PPI is able to obtain more meaningful saliency maps. The training process involves finding saliency maps, masking out the area corresponding to relevant features to obtain negative examples, and then including this in the training pipeline. The training loss adds an L1 regularization to the obtained saliency maps, and also creates randomly masked examples to prevent collapsing to degenerate solutions.  In order to compute (better?) saliency maps faster, the authors also introduce a technique they call Weight Back Propagation (WBP). The idea behind WBP is quite simple and intuitive, however I do have some questions about the approach that I've outlined in the next section.

Overall, I think the paper presents an interesting idea, and shows state-of-the-art results in useful and practical settings. I think the paper should be of interest to a broad range of audience thinking about fundamental problems in deep learning (classification, domain generalization, interpretability), and also practitioners who work with real world data where these topics are of importance. Most claims in the paper are well supported by data, however, please find a few questions/comments I have about the next section

**Broader Impact Concerns:**

No ethical implications in my opinion. However, I think the claim here that this method finds relevant features should be toned down. It may be fair to say that the method finds relevant features in the dataset where they verified it domain experts, but a generic statement might be misleading.

**Requested Changes:**

Please refer to the weakness section. Answering 1, 2 and 4 is critical for securing acceptance, and answering 3 and 5 can significantly strengthen the work

**Strengths And Weaknesses:**

Strengths:

1. The main idea of using saliency maps to mask out parts of image (and combining it relevant regularization scheme and training strategy) is quite interesting and intuitive. The authors also experimentally show this improves performance across a variety of benchmark and real datasets, and that it improves cross domain generalization in a medical dataset.
2. The idea behind WBP is well motivated (although please find some questions below), and it seems to do better than other methods authors compared against to obtain saliency maps.
3. The experiments section works with datasets from different domains and different setting (low amount of data etc.), which is greatly appreciated.
4. The paper is well written and easy to follow. The writing in concise and clear.

Comments/Weakness:

1. **WBP**

WBP looks extremely similar to the algorithm introduced in [Mohan. et. al (2020)](https://openreview.net/pdf?id=HJlSmC4FPS) (see Section 6.1 and equation 5) to compute saliency maps for denoising. Mohan  et. al (2020) is for image denoising (similar idea in [Sheth et. al.](https://arxiv.org/pdf/2011.15045.pdf) for video denoising), and this is for classification, so it's a completely different problem, but the core idea, in my understanding is the same. In Mohan et. al. (2020), they work with bias-free networks (i.e, they removed bias/linear additive terms from each convolution layers, and also make the batch normalization layers linear instead of additive by removing mean subtraction step and the additive learned parameter). Making network bias-free ensures that the first-order Taylor approximation of the network is linear (and not affine), and hence the rows of the matrix can be directly interpreted as filters.

Equation 6 in the paper is exactly doing this - making a local linear approximation of the function, and interpreting the rows of the matrix as filters/saliency. However, it seems like the bias-terms in the convolution layers and batch norm layers are not accounted for? Can the authors clarify on this? For example, in a simple case of a one layered network, where the output is (y = Ax + b), why would eq. (6) make sense?

2. **Details of OCT-GA dataset**. Since this is a new dataset, it might make sense to include some more description on what is the data and task looks like. The only information that Section 4.2 provides now is that the proposed methods works better on this dataset, and does not say anything about what this task even is.

3. **Adversarial Positive Contrasts**. I understand the motivation to prevent overfitting to the intervention. (a) Have the authors verified experimentally that overfitting is indeed a problem here? Further, why do these examples have to generated using saliency maps of other examples? For example, can I just not randomly mask out a patch (instead of obtaining this mask using saliency map of x_j) and still achieve a similar effect? This could potentially speed up training quite a bit. Such controlled experiments would help understand how different components are contributing better.

4. **Disentagling training technique from WRB**. I consider WRB and the training technique involving masking out casual features as two independent contributions here (although they work together), and I find it hard to understand from the current results the contribution of each component. These are some of the questions I'm specifically interested in:
* (a) **Does PPI training improve saliency detection overall?** For example, assume that we have two versions of the network, one trained in a generic way, and another with PPI (maybe with WRB during training(. If I look at the saliency map generated by grad cam or LRP on both of these networks, do I see that the network trained with PPI is better? Can this be quantified using metrics in Fig 4.

* (b) **How much does PPI depend on the method used for obtaining saliency maps during training?** Can WRB be replaced by grad cam or LRP during training, and does it give similar performance?

* (c) **Performance of vanilla WBP when compared to other techniques**. Is Fig 3 (right) already doing this? I couldn't tell what exactly the setting was here. I would like to take regular network, and compare the performance of WRB with other techniques, and visualize the saliency maps and compute metrics.

5. Authors state in the paper that when data is low, networks tends to hang on to spurious correlation more, and PPI can help better in such settings. Can authors create a controlled setting where the number of data points can be varied, and quantity this statement?

---

> ### Author Response · Authors · 2022-09-26
> **Reply to Reviewer x2bF**
>
> Thanks to the reviewer for the valuable feedback. We appreciate that the reviewer recognizes that our method is interesting and intuitive, the experiments are illustrative of improvement and effective in different settings, the proposed WBP is well motivated, and the content is clear. We will address the weaknesses one by one, and comment on the requested changes.
>
> ### Weakness 1 WBP
> The reviewer is right in that for a linear network (simplest case a single linear layer) equation 5 in the referred work and equation 6 in the paper are exactly the same. This is because both are interpreting an inverse to the linear system. In the referred work, bias is removed from each layer and hence equation 5 can be applied through the network. In the paper, however, WBP is applicable to all networks (linear, non-linear, biased, and bias-free networks). An alternative interpretation should be that WBP analytically computes a weight for each input pixel that the inner product of the weights and the input pixels corresponds to the total contribution of the prediction class. WBP only computes the weight not the bias simply because in each layer of the network different weights are applied to the input variables, and bias is applied equally to all variables. WBP investigates how each input pixel interacts with the layer weight from the input layer to the prediction layer. For non-linear activations, WBP treats the layer as a scaling function. In short, WBP is not approximating a linear inverse of each layer in a network, it is evaluating how much weight each layer applies to the input variable so that the weight at the input layer multiplied by the input pixel value represents the contribution of that input pixel to a specific prediction.
>
> ### Weakness 2 OCT-GA dataset
> We include more details on the OCT-GA dataset in the supplementary section E. Please inform the authors if more information is needed for the dataset.
>
> ### Weakness 3
> The reason for including the adversarial positive contrasts is that this encourages the model to be robust to the interventions. We can verify the benefit (also indication of overfitting) by including an ablation study on all loss terms (requested by another reviewer too). However, if only randomly generated masks are used as interventions, the network can detect the random masks as an intervention and therefore be robust to this intervention which contradicts the point of introducing the adversarial positive contrast. In terms of computation time, since the masks are already generated during the backpropagation, a random shift may cost the same time as computing randomly generated masks.
>
> ### Weakness 4 a
> In Figure 5, the improvement of PPI on the saliency maps is visually presented and the explanations are more clinically relevant to the abnormalities. We can include a selection metric of PPI on CUB and visually present it in comparison with Figure 4a.
>
> ### Weakness 4b
> For the results presented in Table 2, the $PPI_*$ represents the PPI framework with a specific saliency mapping method. $PPI_{Gradient}$ is using a gradient as the saliency mapping method to train the network with the proposed loss function.
>
>
> ### Weakness 4 c
> In Figure 3 the benefit of WBP focusing more on the target instead of some spurious cues is visually presented in comparison with other techniques. In Figure 4, the setting is just to compute the saliency maps with different techniques and delete the input pixels from most important to the less important, and compute the reduction in prediction score. So Figure 4 along with Figure 3 addresses the benefit of vanilla WBP.
>
> ### Weakness 5
> In the paper, it is stated that the effect of spurious correlation is particularly notable in small-sample-size (weak supervision) scenarios or when the sources of non-informative variation are overwhelming, and the generalization issue worsens with a smaller training sample size. This is because models trained on small data sizes may make predictions on the cues only presented in the small-size dataset, and these cues may not be generalized well in a larger dataset. In Table 3, the results on small-sized medical data from different hospitals illustrate that hospital D picks up predictive cues from the in-distribution data but these cues do not generalize well to datasets from other hospitals. Applying PPI during training on the dataset from hospital D not only improves the in-distribution dataset but also learns more generalized cues to improve performances on other datasets. Additional experiments emphasize the relation between the size of training data and the benefit of PPI is not essential since that PPI can address the spurious correlations in both large-sized data (natural images) and small-sized data (medical data).
>
> # Requested changes
> Please refer to the Weakness sections for the requested changes.

---

### Review · Reviewer_5Zgf · 2022-09-18

**Summary Of Contributions:**

The main contribution of the paper is a new learning algorithm called PPI which ensures that the network makes predictions based on causal relevance, not spurious correlations.
The experiments on multiple datasets demonstrate that PPI achieves better classification and interpretability performance.
PPI enforces the network to predict a wrong class when the object of interest is masked while predicting the correct class when random background locations are masked.
Masking is done based on the extracted saliency points during training.
This encourages the network to focus on the object of interest, not the spurious correlations that may arise in the background.
Also, the paper proposes a new salience mapping method called WBP.
The experiments performed on multiple datasets demonstrate that PPI with the proposed mapping method achieves superior method compared to the other saliency mapping schemes.



**Broader Impact Concerns:**

I think talking about the limitations and failure cases of the proposed method is crucial and it is missing in the paper. Such algorithms are intended to be used in safety-critical applications such as medical imaging to improve the trustworthiness of automated systems. So, one would need to know when algorithms may fail to trust them.

**Requested Changes:**

I would request the authors to address the points in the weaknesses section above.

**Strengths And Weaknesses:**

**Strengths**
- The idea of using saliency maps during training for extracting background and foreground masks and using this information to guide the training such that the network focus on the causally relevant features is quite interesting.
- The method description is quite clear, although there are unclear parts and misuse of some terminologies.
- The paper experiments on multiple datasets, showing clear performance improvement over the existing methods.

**Weaknesses**
- The paper proposes multiple loss functions during training; however, the contribution of each term is not shown in an ablation study.
- The terminology "contrastive" is a bit misleading. The main idea in conventional contrastive learning is to train a network such that the representations of positive samples are pulled together while they are pushed away from negative samples. There is no such mechanism in the contrastive loss prosed in the paper. Is there any relation between the proposed loss and the conventional contrastive loss used in the literature.
- Limitations of the proposed method are not discussed. For example, when choosing the random mask for positive contrast, the object of interest in x_j may correspond to the location of the object of interest in x_i. In this case, the foreground object will be masked in x_i, and this will be equal to the negative contrast case. How would such cases affect training?

---

> ### Author Response · Authors · 2022-09-26
> **Reply to Reviewer 5Zgf**
>
> Thanks to the reviewer for the valuable feedback. We appreciate that the reviewer recognizes that our method is interesting, the content is clear for the most part, and the experiments are illustrative of improvement. We will address the weaknesses one by one, and comment on the requested changes.
>
> ## Weakness 1
> The reviewer suggests evaluating the importance of each loss term in the loss function. We agree that each term is proposed for a specific reason. We will include an ablation study on the loss function.
>
> ## Weakness 2
> The reviewer is right in that recent popular work on ‘contrastive learning’ trains the model to push positive representations closer and negative representations further by contrasting examples. However, contrastive by itself means contrasting examples. In the proposed framework, a (positive) example is generated by placing a random mask on the original image that would not change the semantic meaning of the image, and another (negative) example is generated by masking out the causal pixels in the original image. This mechanism shares similarities with the core idea referred to by the reviewer. We will include a clearer reference in the Contrastive Learning section.
>
> ## Weakness 3
> The reviewer points out that there is a possibility that different target objects generate similarly located masks and contrasting these similar masks may cause contradicting signals during the training. In natural images, we mask out important pixels for contrasting. Even though different objects may share similar locations, pixels are less likely to overlap. The soft-masking function T() is applied to further filter less important pixels and therefore making overlapping pixels even less likely. On medical images, when the box masking variation is applied, it is more likely to overlap just as the reviewer conjectures. However, \sigma applied to control the size of the box also determines the optimal size that ensures less overlap between masks in a mini-training batch. Other limitations of the proposed method include longer computation time and hyperparameter tuning. We will include a section to mention these limitations and how we lessen these limiting effects during the implementation.
>
> # Requested changes
> See Weaknesses 1,2,3 for requested changes.

---

### Decision · Action_Editors · 2022-10-24

**Recommendation:** Reject

**Comment:**

While all reviewers see some merits in this submission, several concerns about the clarity and presentation of this submission have been raised. For example,  Reviewer 6PLv raised the issue of similarity between WBP and the Jacobian / Gradient, as well as the technical correctness of Section 3.2. Reviewer MiEG also shared concerns on claims and evidence (see the author-reviewer discussion thread). The authors' response and rebuttal did not fully address these concerns. Most reviewers agree that the current version requires significant and major revision with more direct and convincing claims and evidence. Therefore, I recommend rejecting the current version.

**Audience:**

Yes. The topic falls into the general interest of TMLR's audience

**Claims And Evidence:**

Some claims are not entirely justified. For example,  Reviewer 6PLv raised the concern of similarity between WBP and the Jacobian / Gradient, as well as the technical correctness of Section 3.2. Reviewer MiEG also shared concerns on claims and evidence.